# Exploring Monthly Variation of Gait Asymmetry During In-Hand Trot in Thoroughbred Racehorses in Race Training

**DOI:** 10.3390/ani15162449

**Published:** 2025-08-20

**Authors:** Thilo Pfau, Bronte Forbes, Fernanda Sepulveda-Caviedes, Zoe Chan, Renate Weller

**Affiliations:** 1Faculty of Kinesiology, University of Calgary, Calgary, AB T2N 1N4, Canada; zoe.chan1@ucalgary.ca; 2Faculty of Veterinary Medicine, University of Calgary, Calgary, AB T2N 1N4, Canada; renate.weller@ucalgary.ca; 3Hong Kong Jockey Club, Hong Kong, China; 4Department of Clinical Science and Services, The Royal Veterinary College, Hatfield AL9 7TA, UK

**Keywords:** Thoroughbred, movement symmetry, monthly variation

## Abstract

Racehorses routinely undergo strenuous training exercise. Head and pelvic movement symmetry are established measures for detecting clinically relevant movement abnormalities (“lameness”) that indicate that one or multiple limbs show a reduced weight bearing and/or pushoff force. We explore how often, when measuring movement symmetry with inertial sensors at monthly intervals, switches between left- and right-sided asymmetry are observed exceeding three different thresholds: (1) left–right switches around perfect symmetry; (2) switches indicative of left- or right-sided lameness; and (3) switches outside published repeatability values for racing Thoroughbreds. The frequency of left–right switches was compared between the three thresholds. A total of 256 Thoroughbreds in race training contributed multiple data sets at monthly intervals, and absolute difference values for head and pelvic movement symmetry were compared to three pre-defined thresholds. Monthly variation values exceeded the published daily and weekly variation. Left–right switches, compared to perfectly symmetrical movement, were found frequently (approx. 30% of the time). Applying clinical lameness thresholds significantly (*p* < 0.001) reduced these switches to 4 to 17%, and utilizing daily repeatability values further significantly (*p* < 0.001) reduced the switch frequency to 7% or below. Racehorses in training regularly switch between left- or right-sided movement symmetries. However, they rarely switch between more pronounced asymmetries defined based on daily variations. Further studies should investigate reasons for these rare switches.

## 1. Introduction

Epidemiological studies in Thoroughbred racehorses in training indicate that high speed exercise is one of the risk factors for the development of, as well as for the prevention of, musculoskeletal injuries [1,2,3,4,5]. High-speed gallop should be considered a strenuous exercise, with horses producing forelimb peak forces of up to 2.5 times their body weight [6]. Gallop is an asymmetrical gait [7], and small differences in movement symmetry have been found between groups of racehorses training and racing in opposite pre-dominant directions (clockwise or anti-clockwise), generally indicating reduced force production with the limb pre-dominantly on the outside of the circle [8]. However, with values in the order of 3 mm [8], the documented racing direction-specific differences are small compared to the “normal variation” (between days or weeks) previously measured in a small group of Thoroughbreds in training [9]. In Standardbred trotters, utilizing a symmetrical gait while racing, there is evidence of increased head movement asymmetry with increasing training intensity (speed and/or uphill) [10], and increased movement asymmetry has been linked to poor lifetime earnings [11].

Head and pelvic movement symmetry are frequently used in clinical lameness examinations and have been shown to be associated with force differences between contra-lateral limbs [12,13] in accordance with Newtonian mechanics. It is this association between movement symmetry and force symmetry that indicates that Thoroughbreds in race training show evidence of a variation in force production between contra-lateral limbs across days (and weeks) [9]. They also show small movement asymmetry changes in reaction to pre-dominant training and racing direction [8]. We are interested in further evaluating whether such quantitative movement symmetry data might be useful for exploring the onset of impending overuse injuries, for example, by identifying horses that may benefit from further veterinary assessments. Importantly, this task is different from clinical lameness examinations; i.e., it is different from identifying the lame(st) limb in a horse with a pre-conceived notion that “something is wrong”, and further clinical examinations need to be conducted. In the context of lameness examinations, the measurement of (small amounts of) movement asymmetry is used for identifying a limb that is contributing less to force production in comparison to its contra-lateral counterpart. In the context of detecting impending injuries, the task is to identify horses that are likely to develop an injury in the future and, for example, have been shown to demonstrate comparatively small adaptations to their “in-race” gallop stride parameters [14]. We are asking the question whether longer term (monthly) variations in movement symmetry might be useful in this context by quantifying the frequency of switches between left- and right-sided movement asymmetries as a “pre-screening” tool for identifying horses that might benefit from further veterinary assessments. We implicitly assume that switches between more pronounced movement asymmetries, which have been related to larger force asymmetries [12,13], are more likely related to an attempt of “unloading” one of the limbs, and we postulate that such horses might benefit from further veterinary inspections.

The aim of the present study was to provide a first exploration of a long-term database collected from Thoroughbred racehorses in active race training, for which vertical head and pelvic movement symmetry had been quantified during in-hand trot. We hypothesize that changes between repeated in-hand gait assessments conducted in trot at monthly intervals in Thoroughbreds in active race training regularly exceed previously established clinical lameness thresholds [15]. We also hypothesize that a smaller number of horses would show switches between left- and right-sided asymmetry when applying previously established “lameness” thresholds [15], and a further reduced number of horses would show left–right switches when using discipline-specific “repeatability” values as cutoff values [9].

## 2. Materials and Methods

Data collection for this study had been approved by the Royal Veterinary College Clinical Research Ethics Review Board (CRERB) with the unique reference number URN 2013-1238. Informed consent for inclusion in this study was provided by the trainers of the horses as agents.

### 2.1. Horses

A total of 286 Thoroughbreds in training at Singapore Turf Club between November 2014 and May 2016 were chosen to participate in a longitudinal study aiming to perform movement symmetry assessments at monthly intervals during in-hand, straight-line trot. Data collection was consistently performed after morning exercise, approximately between 10 AM and 3 PM.

All horses included in the present study were Thoroughbreds within the standard breed height range of 15 to 16.1 hands (range = 152.4–163.6 cm) and comprised 249 geldings, 6 entire males, 28 females, and 3 of unknown sex.

### 2.2. Gait Assessment

Movement symmetry and range of motion were calculated from vertical displacement data of four inertial measurement units (Xsens, MTw, gyroscope range: +/−2000 degrees/s; accelerometer range+/−160 m/s^2^; magnetometer range +/−1.9 Gauss; sample rate ≥ 60 Hz per individual data channel). One sensor was attached to the head piece (or head collar) over the poll with custom pads and secured with Velcro straps. The three remaining sensors were attached over the pelvic area with custom Velcro pads and double-sided tape: one sensor over the midline of the pelvis (sacrum: between the two tubera sacrale) and one sensor each over the craniodorsal edge of the left and right tuber coxae.

Horse assessments were scheduled at approximately monthly intervals over the entire data collection period unless horses were unavailable, for example, due to the racing schedule. Horses being removed from active training, being sold to a different training jurisdiction, or unable to be trotted up, for example when being considered lame by the trainer or the veterinarian conducting the gait assessment (F.S.C.), were excluded from data analysis. Horses were excluded from further data collection when after the first assessment they were considered “unsafe” either by the handlers or the veterinarian (F.S.C.).

### 2.3. Movement Variables

Movement symmetry and range of motion values were calculated following published methods [16,17]. In brief, continuous data streams for tri-axial acceleration, tri-axial rate of turn, and tri-axial magnetic field strength (update rate ≥ 60 Hz [18]) were recorded by manually starting and stopping data recording (via Xsens MTManager software, v4.8) during in-hand, straight-line trot on hard ground. Acceleration data were rotated into a horse and gravity-based reference frame using Euler angles (provided by the proprietary Xsens sensor fusion algorithm) of each individual sensor. Rotated acceleration data were then double-integrated to velocity and displacement. Finally, continuous data were segmented into individual stride cycles [19] and median values for vertical movement symmetry and vertical range of motion calculated across all stride cycles of each gait assessment.

Movement symmetry variables were defined as the difference between the two vertical minima (head: HDmin, midline pelvis: PDmin), between the two vertical maxima (head: HDmax, midline pelvis PDmax), and between the two vertical upward movement amplitudes (head: HDup, midline pelvis: PDup) of each displacement trace for each individual stride cycle. The hip hike difference (HHD) was calculated from vertical movement amplitudes of the left and right tuber coxae. The following sign convention was used across all movement symmetry variables: negative values were assigned to displacement traces representing movement patterns typically observed in left lame horses and positive values to displacement traces representing movement patterns typically observed in right lame horses. For example, a minimum difference HDmin (or PDmin) representing a trace where the head (or midline pelvis) remains at a higher position during left fore (or left hind) stance would be assigned a negative value based on the association with reduced peak force production with the left front (or hind) limb [12,13]. For a trace where the maximum or upward movement amplitude was smaller after right fore (or hind) limb stance, HDmax or HDup (or PDmax or PDup) were assigned positive values based on the association with reduced force (impulse) production with the right front (or hind) limb [12,13]. HHD was negative for horses showing increased upward movement amplitude of the left tuber coxae during and after right hind stance [20].

Vertical range of motion was calculated for each stride cycle from the overall minimum value subtracted from the overall maximum value of vertical displacement resulting in an unsigned (positive) range of motion parameter. All movement symmetry and range of motion parameters were expressed in millimetres.

### 2.4. Data Analysis

A total of 256 of the original 286 horses contributed at least two gait analysis data sets to our database and were consequently included in the calculations. Table 1 lists the number of horses contributing data sets to the overall database.

The number of days lapsed between consecutive assessments were calculated based on the date of each assessment and a histogram plotted to illustrate the distribution of these data as well as descriptive statistics calculated (minimum, 5th, 10th, 25th, 50th, 75th, 90th, 95th percentile, and maximum).

#### 2.4.1. Movement Symmetry Distribution

Head and pelvic movement symmetry data of all 256 horses included in the final analysis, i.e., the 1632 gait assessments of horses contributing at least two movement symmetry data sets, were categorized to illustrate the distribution of movement asymmetries found in the specific cohort of Thoroughbred horses randomly chosen from horses in race training at Singapore Turf Club. Specifically, for each movement symmetry parameter, the number of gait assessments presenting evidence of the presence of left or right asymmetrical or symmetrical movement were quantified as follows. Thresholds of 8 mm for head movement and 4 mm for pelvic movement were defined based on the values of 6 mm for head movement and 3 mm for pelvic movement [15] corrected based on equations presented in [21]. Values smaller than −8 mm or smaller than −4 mm were labelled as “left”, values exceeding +8 mm or +4 mm as “right”, and the remaining values as “symmetrical”. This procedure was implemented for each of the seven movement symmetry variables separately.

#### 2.4.2. Variability of Gait Symmetry and Range of Motion at Monthly Intervals

In order to compare the variability of our longitudinally collected data at monthly intervals to previously quantified variability estimates for daily and weekly repeat gait assessments in Thoroughbred racehorses in training [9], the following calculations were implemented. Absolute values of differences in movement symmetry and range of motion variables were calculated between consecutive gait assessments, resulting in 1376 differential data sets from the 256 horses with more than one gait assessment. We also calculated the same differences only for a subset of the data. This subset consisted of the repeat assessments that lay within the middle 90% of the distribution of the days lapsed, i.e., repeat assessments conducted within 26 to 77 days of the previous assessment (see Table 2 for the descriptive statistics of days lapsed). This was performed to exclude repeat assessments from analyses that were conducted at very long or very short time intervals between repeats—for example, horses returning to training from a long period out of training.

#### 2.4.3. Frequency of Switches Between Left- and Right-Sided Asymmetry

Our main study interest was the investigation of the number of times where two consecutive gait analysis measurements were found to show a “switch” in the direction of asymmetry shown. Three different metrics were used and applied to each movement symmetry measure separately.

The first calculation was performed on quantitative, directional movement symmetry data and consisted of the number of times when there was a switch from a negative (left asymmetrical) value to a positive (right asymmetrical) value or vice versa; i.e., any deviation from “perfect symmetry” was taken into account when counting the frequency of switching between left- and right-sided movement.

The second and third calculations were performed on categorical data, making use of the three different asymmetry categories (“left”, “right”, and “symmetrical”, see “Movement Symmetry Distribution” Section) defined using thresholds of 8 mm for head movement and 4 mm for pelvic movement (“lameness thresholds”) or values of 16.5 mm for head movement and 10.5 mm for pelvic movement (previously established “Thoroughbred repeatability values”).

Based on the categorical data, direct switches between the categories “left” and “right” were counted between consecutive measurements; i.e., only switches were counted when the horse showed very clear evidence of switching preference between contra-lateral limbs.

Switch frequencies, i.e., the proportion of differences between repeat assessments that showed evidence of a switch from left- to right-sided asymmetry, were compared with a paired samples proportion test applying both Mid-*p* Adjusted Binomial and McNemar tests, with a level of significance of *p* < 0.0167 considered indicative of a pairwise statistically significant difference between the three threshold levels (“perfect symmetry” vs. “lameness threshold”; “perfect symmetry vs. “repeatability threshold”; and “lameness threshold” vs. “repeatability threshold”).

## 3. Results

### 3.1. Movement Symmetry Characteristics of Thoroughbred Horses in Training Included in This Study

Table 3 presents descriptive statistics for the three head movement symmetry variables, the four pelvic movement symmetry variables, and the two range of motion variables for all measurements conducted with the 256 study horses that contributed at least two gait assessments to the database. Median values are generally small with values between −2 mm and +3 mm for the symmetry variables. There is considerable spread within the data set, indicated by the 5th percentile values varying between −39 mm and −16 mm for movement symmetry and the 95th percentile values varying between 16 mm and 39 mm.

Using guideline “lameness” values for categorizing movement symmetry data into “left asymmetrical”, “symmetrical”, and “right asymmetrical” categories, between 20 and 58% of values for each movement symmetry parameter indicate symmetrical movement (Table 4). Between 18% (HDmax) and 44% (HHD) of values indicate left asymmetrical movement for each movement symmetry variable. Between 24% (HDmax) and 39% (PDup) indicate right asymmetrical movement.

### 3.2. Variability of Repeat Gait Analysis Measurements

Figure 1 and Figure 2 present histograms for absolute values of differences between repeat measurements of head and pelvic movement symmetry measures from N = 256 Thoroughbred racehorses in active race training. For head movement, the 90th percentile of absolute movement symmetry differences between repeats were found to lie between 20 mm for HDmax and 37 mm for HDup (Table 5). For pelvic movement, the 90th percentile of absolute movement symmetry differences between repeats were found to lie between 13 mm for PDmin and 24 mm for PDup for midline pelvic movement and 30 mm for tuber coxae movement (HHD) (Table 5). The corresponding values for the 95th percentile are 23 mm (HDmax) and 49 mm (HDup) for head movement and 17 mm (PDmin), 30 mm (PDup), and 38 mm (HHD) for pelvic movement.

In addition, Table 5 also lists values for daily and weekly repeat measurement differences reported previously [9]. With the exception of the vertical range of motion of the pelvis (PROM), the previously reported daily and weekly repeat measurements show smaller values for the 90th and 95th percentile of absolute differences between repeat measurements compared to the equivalent percentiles measured at monthly intervals in the present study.

Table 6, Table 7 and Table 8 provide an overview of the number of repeat assessments associated with evidence that horses have switched between left- or right-sided movement symmetry.

Based on Table 6, between approximately 27% and 31% of repeat measurements indicate a switch from left-sided (negative value of the associated parameter) to right-sided (positive value) asymmetry for pelvic movement. For head movement, this varies between 30% and 34%. The percentage of measurements showing evidence of a switch in any of the pelvic symmetry variables is 56%, and for switch in any of the head symmetry variables this percentage is 59%.

The number of times when a horse shows a switch between different asymmetry categories (from left to right or vice versa), when different threshold values are used, are presented in Table 7 and Table 8. When considering “lameness thresholds” with values outside +/−8 mm for head movement and +/−4 mm for pelvic movement as left- or right-sided asymmetry, between 4 and 11% of repeat values indicate left/right switches for head movement and between 7 and 17% for pelvic movement (Table 7). Using these definitions, the percentage of measurements showing evidence of a switch in any of the pelvic symmetry variables is 28%, and for a switch in any of the head symmetry variables this percentage is 14%.

More stringent definitions of left- or right-sided asymmetry based on “daily repeatability values” using +/−16.5 mm for head movement and +/−10.5 mm for pelvic movement, between 0.3 and 3.6% of repeat measurements for head movement and between 0.6 and 7.0% for pelvic movement, show evidence of a switch between left- and right-side asymmetry (Table 8). Using these definitions, the percentage of measurements showing evidence of a switch in any of the pelvic symmetry variables is 8.3%, and for a switch in any of the head symmetry variables this percentage is 3.9%.

Both mid-*p* adjusted Binomial and McNemar test statistics indicated statistically significant differences (all *p* < 0.001) for all movement symmetry variables (HDmin, HDmax, HDup, PDmin, PDmax, PDup, HHD) for all pairwise tests between the three threshold levels (“perfect symmetry”, “lameness threshold”, and “repeatability threshold”). In general, differences in “switch proportions” (the percentage of repeats indicative of a left-right switch) were largest for “perfect symmetry” vs. “repeatability threshold”, followed by “perfect symmetry” vs. “lameness threshold” and then “lameness threshold” vs. “repeatability threshold”.

## 4. Discussion

In the present study movement symmetry of vertical head and pelvic displacement was assessed at monthly intervals in a cohort of Thoroughbred racehorses in active race training at a single facility. Horses were training and racing in an anti-clockwise direction. In contrast to a previous study at the same facilities, which investigated daily and weekly variability in a limited number of horses [9], our current study has focused on a larger cohort of horses and “longer term” variability of movement symmetry over multiple consecutive measurements conducted at monthly intervals.

First, statistics on absolute differences in movement symmetry values were calculated for a direct comparison with previously established values for daily and weekly absolute differences [9]. In alignment with our hypothesis, monthly variability values were found to exceed these metrics [9] for all head movement variables. Interestingly, for pelvic movement, the monthly variability of symmetry variables exceeded the daily and weekly values; however, the pelvic range of motion did not. This is a somewhat curious finding. We speculate that this may reflect differences in “functional mechanics” between the pelvic limbs and the thoracic limbs in relation to upper body movement and specifically different pelvic roll patterns that have been documented, albeit in a very different group of horses (non-lame Warmblood horses) [22]. The existence of different pelvic roll patterns might also explain why estimating tuber coxae movement from a single inertial sensor mounted over the midline of the pelvis is associated with variable success [23]. If, for example, the different roll patterns are related to a different timing of gluteal muscle contractions over a stride cycle, this could influence the relationship between the movement recorded from a sensor mounted over the tubera sacrale and the vertical movement of the left and right tubera coxae.

As the main focus of this study, we investigated how frequently horses show evidence of a switch in the direction of upper body movement symmetry at monthly intervals. Three different types of switches were considered: (1) switching between left- and right-sided asymmetry regardless of the amount of asymmetry, i.e., deviations from “perfect symmetry”; (2) switching between left- and right-sided asymmetry when categorized by previously established “lameness thresholds”; and (3) switching between left- and right-sided asymmetry when categorized by previously established Thoroughbred-specific “repeatability values”.

When considering any switch from a negative value to a positive value or vice versa between repeat assessments, independent of the movement parameter, in approximately 27 to 34% of assessments, i.e., approximately every third to forth month, a horse would show evidence of a left–right switch. When considering the frequency of switching of any of the three head (or four pelvic) movement parameters, a switch occurs in over half (59%) of the repeat measurements (or 56% for pelvic movement).

When only counting switches between values that would be considered as a switch between a left and a right lameness (or vice versa) based on clinical “lameness thresholds”, only between 4 and 11% for the different head movement variables and between 7 and 17% for the different pelvic movement variables show evidence of a switch. It should be noted that in the present study we chose thresholds of 8 mm for head movement and 4 mm for pelvic movement, which are derived from published values [15] adapted by published correction equations for the sensor system utilized here [21]. Other systems might require further adaptations of these values.

The number of left to right switches further drops to between 0.3 and 3.6% for the different head movement measures and to between 0.6 and 7.0% for the different midline pelvic symmetry measures and to 3.4% for hip hike when applying Thoroughbred-specific repeatability values [9].

It hence appears that when measuring right-sided asymmetry outside “clinical” or “repeatability” thresholds that such a horse is unlikely (“clinical threshold”) or very unlikely (“repeatability threshold”) to show evidence of left-sided asymmetry outside the same pre-defined threshold in its monthly follow ups. Given that these more “severe” changes appear to be fairly infrequent, it might be feasible for future studies to focus on these horses, in particular if monthly assessments could be conducted with a markerless tracking technique, for example, only requiring a standard SmartPhone [24] or possibly relying on features extracted from “in-race” sensors extracting movement symmetry values from pre-race warmup or post-race cool-down periods. Thresholds for such systems need to be further investigated.

In contrast to these more dramatic changes between left and right asymmetrical patterns, the fact that 30% of repeat measurements for individual variables (or up to 59% for any observed switch) indicate a switch between left- and right-sided asymmetry when using switches around 0 mm (“perfect symmetry”) confirms that small movement asymmetries are generally not reliable indicators of a consistent asymmetrical movement pattern. Based on comparisons to force platform data, head and pelvic movement symmetries are however reliable indicators of force differences between contra-lateral limbs [12,13]; however, the small magnitudes of movement symmetry variables also indicate that these movement asymmetries are related to small force differences. Hence, it seems that Thoroughbred racehorses in active race training show evidence of frequent small shifts in the force between contra-lateral limbs.

No matter which metric is being used, establishing what is “normal” in terms of repeatability of gait analysis measurements appears important. The median values for monthly absolute differences in the present study (head: 7 to 13 mm; pelvis: 5 to 9 mm) are similar to the confidence intervals for differences between two consecutive measurements reported previously for a mixture of non-lame and lame horses with a different inertial sensor system (head: approximately 9 mm; pelvis: 4 to 5 mm) [25] and also for between-measurement variation for a 3D optical motion capture system (head: 12 to 20 mm; pelvis: 4 to 7 mm) [26]. However, when considering the ranges containing 90% or 95% of the monthly differences reported here, the values in our cohort of Thoroughbred racehorses in active race training approximately double or triple compared to the median values.

Our findings indicate that there are large variations between individual horses, and, in particular, in the context of preventative measurements or following horses over the course of a rehabilitation regimen, it would seem advisable to undertake further studies investigating possible reasons for this variability in different groups of horses, for example, as a function of breed, age, sex, and discipline. This also appears of interest for additional movement-related parameters, such as thoraco-lumbo-sacral ranges of motion that might be of interest in horses with a poor performance. Accuracy, precision and repeatability of spinal movement characteristics achieved with inertial sensors or motion capture are encouraging [17,27,28]. In particular, inertial sensors provide the necessary flexibility to undertake measurements “in the field” and—in analogy to 3D optical motion capture—allow for simultaneous quantification of movement symmetry and multi-dimensional thoraco-lumbo-sacral range of motion [29]. We consider this—together with the use of surface electromyography of limb, core, and back muscles under a variety of conditions [30,31,32,33,34,35]—a promising approach for enhancing our knowledge about the association between movement symmetry and lameness in the context of, for example, poor performance examinations and rehabilitation regimens.

### Study Limitations

A clear limitation of the present study is the limited amount of additional information available about each horse over and above the gait data collected. In particular, it was not possible to access the clinical records of the horses. Hence, no attempt was made in this study to utilize the longitudinal nature of the monthly repeat assessments, such as, for example, in recent studies utilizing “in-race” stride parameter data for retro- or prospective predictions of horses at risk of injury [14,36]. In contrast, our explorations were focused on quantifying the switch frequency between left- and right-sided movement asymmetry as a means of estimating how many horses would be identified as potential candidates for further veterinary assessments based on different levels of left–right switches. As such, no attempt was made to implement longitudinal modelling, which is essential for retro- or prospective data-driven injury prediction [14,36].

At the time of data collection, notes had been made about any information that the trainers or stable staff deemed important. However, given the large number of assessments and the lack of a standardized form for categorizing any additional piece of information, together with a large variation in the number of gait assessments available from each horse, our present study is restricted to reporting descriptive statistics about the variability of monthly gait assessments conducted in a group of Thoroughbred racehorses in active race training at a single facility. The study focus was put firmly on the “switch frequency” and the influence of different threshold values in an attempt to provide a practical assessment of how many horses would potentially benefit from additional veterinary assessments based on “large variations” in the gait asymmetry between two consecutive gait assessments as a “pre-screening” tool.

It would appear essential to further investigate how horse-, training-, racing-, and medical history-related factors, such as the age, breed, sex, presence and/or mileage of high-speed exercise, previous racing success, or previous injuries, might influence monthly gait variability. This is beyond the focus of the current study, in particular since no consistent medical or training records were available from the horses. Amongst others, these factors have been investigated as risk factors for fractures in flat racehorses [1,2,3,4,5,37,38], and the accumulative overuse nature of many racehorse fractures may influence day-to-day as well as month-to-month variability of in-hand gait symmetry given its fundamental association with load distribution [12,13]. This necessitates longitudinal modelling approaches, such as those implemented previously [14,36], when aiming for an “autonomous”, data-driven injury prediction approach and emphasizing the collection of additional training and injury data.

## 5. Conclusions

When conducting inertial sensor-based head and pelvic movement symmetry assessments at monthly intervals, racehorses in training regularly show evidence of switches between small left- or right-sided movement symmetries. However, when using previously established cohort-specific repeatability values, switches between more exacerbated left- and right-sided movement symmetries are comparatively rare: in our database of 1376 monthly repeat assessments, this happened between 0.3 and 3.6% for individual head movement symmetry parameters and between 0.6 and 7.0% for individual pelvic movement symmetry parameters. It would seem interesting to conduct similar studies in different groups of horses (breed, age, sex, discipline), investigating potential links with training intensity and to further investigate whether there are specific reasons for the more exacerbated switches—for example by subjecting such horses to further veterinary assessments. Ultimately, a combination of gait asymmetry measurements for pre-screening of horses might help detecting horses at risk of injury through additional veterinary assessments and associated interventions.

## Figures and Tables

**Figure 1 animals-15-02449-f001:**
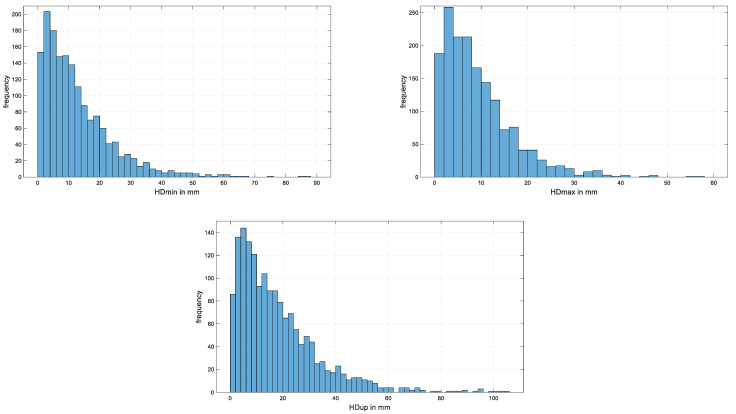
Histograms for absolute values of head movement symmetry measure differences for N = 256 horses contributing at least two data sets. (**Top left**): HDmin; (**Top right**): HDmax; and (**Bottom**): HDup. Horizontal (x) axes are scaled to accommodate the respective maximum values observed; bin width for all histograms: 2 mm.

**Figure 2 animals-15-02449-f002:**
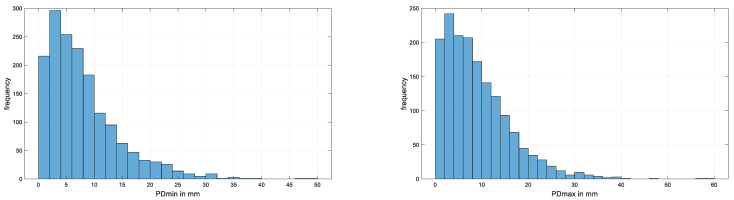
Histograms for absolute values of pelvic movement symmetry measure differences for N = 256 horses contributing at least two data sets. (**Top left**): PDmin; (**Top right**): PDmax; (**Bottom left**): PDup; and (**Bottom right**): HHD. Horizontal (x) axes are scaled to accommodate the respective maximum value observed; bin width for all histograms: 2 mm.

**Table 1 animals-15-02449-t001:** Overview of the number of collected data sets per horse (straight-line, in-hand movement symmetry assessment with inertial sensors) to illustrate the number of horses included in this study. The table shows the number of horses as a function of data sets collected per horse. All 286 horses had at least one data set; 256 horses had at least two data sets collected and hence contributed to the calculation of monthly differences. Two horses had the maximum number of 16 data sets collected over the entire data collection period.

# Data Sets Per Horse	Number of Horses	Accumulative Number of Horses	Accumulative Number of Data Sets
1	30 *	30 *	not included
2	32	62	64
3	40	102	184
4	32	134	312
5	25	159	437
6	18	177	545
7	23	200	706
8	19	219	858
9	15	234	993
10	10	244	1093
11	11	255	1214
12	9	264	1322
13	7	271	1413
14	8	279	1525
15	5	284	1600
16	2	286	1632

* Horses not included in this study due to less than the required two data sets available.

**Table 2 animals-15-02449-t002:** Descriptive statistics for the number of days between consecutive movement symmetry assessments. The median number of days between assessments was 35 days; 5th and 95th percentiles were 26 and 77 days.

Statistic	Days Lapsed
Minimum	5
5th percentile	26
10th percentile	28
25th percentile	29
Median	35
75th percentile	42
90th percentile	63
95th percentile	77
Maximum	315

**Table 3 animals-15-02449-t003:** Descriptive statistics of movement symmetry and range of motion measurements of N = 256 Thoroughbred racehorses in active race training quantified during monthly, straight-line, in-hand trot gait assessments over a period of 16 months. Minimum, 5th, 10th, 25th, median (50th), 75th, 90th, 95th percentile, and maximum values are presented.

	HDmin	HDmax	HDup	HROM	Pdmin	PDmax	PDup	HHD	PROM
Min	−74	−55	−105	28	−49	−56	−79	−101	36
5th	−27	−18	−36	46	−16	−19	−29	−39	53
10th	−18	−13	−25	50	−10	−15	−21	−28	57
25th	−7	−6	−11	57	−5	−8	−11	−15	64
Median	2	1	3	66	1	−1	0	−2	73
75th	11	8	16	75	7	6	10	10	80
90th	20	14	29	84	13	13	20	22	86
95th	26	19	39	91	17	16	27	31	90
Max	86	56	102	125	34	59	85	108	109

**Table 4 animals-15-02449-t004:** Number of gait assessments classified as “left” or “right” asymmetrical or “symmetrical” based on pre-defined “lameness” thresholds of +/−8 mm for head movement and +/−4 mm for pelvic movement. Data from 1632 gait assessment of N = 256 Thoroughbred racehorses for which at least two data sets had been available.

	HDmin	HDmax	HDup	PDmin	PDmax	PDup	HHD
# Left	371 (22.7)	297 (18.2)	448 (27.5)	418 (25.6)	622 (38.1)	628 (38.5)	721 (44.2)
# Sym	761 (46.6)	947 (58.0)	565 (34.6)	647 (39.6)	550 (33.7)	368 (22.5)	326 (20.0)
# Right	500 (30.6)	388 (23.8)	619 (37.9)	567 (34.7)	460 (28.2)	636 (39.0)	585 (35.8)
# Right−# Left	+129	+91	+171	+149	−162	+8	−136

**Table 5 animals-15-02449-t005:** Descriptive statistics for differences in absolute values of head and pelvic movement symmetry and range of motion between monthly repeat assessments in N = 256 Thoroughbred racehorses in training assessed at least twice. Values presented are absolute differences in HDmin, HDmax, HDup, and HROM for head movement and PDmin, PDmax, PDup, HHD, and PROM for pelvic movement. Values in brackets indicate values found to be different when calculating statistics based on repeat assessments between 26 and 77 days lapsed between repeats, representing the 5th and 95th percentile of days lapsed between measurements. There were a total of 1376 repeat assessments and a total of 1252 repeat assessments when limiting data analysis to repeat assessments recorded at intervals between 26 and 77 days ^1,2^.

	HDmin	HDmax	HDup	HROM	PDmin	PDmax	PDup	HHD	PROM
min	0	0	0	0	0	0	0	0	0
5th	1	1	1	1	0	0	1	1	1
10th	2	1	2	1	1	1	1	2	1
25th	4	3	6	3	2	3	4	5	2
50th	9 (8)	7	13	7	5	6	9 (8)	11 (10)	5
75th	15	13	24	13 (12)	9	10	16	19	9
90th	25 (24)	20	37	21 (20)	13	16 (15)	24	30 (29)	13
95th	33 (32)	23	49 (48)	26	17	20 (19)	30	38 (37)	17 (16)
max	89	67	134	76	38 (34)	60	80	105	70 (67)

daily90 *	14	16	NA	12	11	9	NA	12	15
daily95 *	16	20	NA	18	11	11	NA	15	18
weekly90 *	19	18	NA	17	12	13	NA	11	19
weekly95 *	26	22	NA	17	13	18	NA	15	27

* Previously published reference values from [9]. ^1^ The table also includes previously published values for daily and weekly repeat measurements [9], and these are labelled as “daily90” (“daily95”) for the 90th (95th) percentile of daily differences and “weekly90” (“weekly95”) for the 90th (95th) percentile of weekly differences. ^2^ All values are given in millimetres; min: minimum; max: maximum; remaining statistics are percentiles from the 5th to the 95th percentile.

**Table 6 animals-15-02449-t006:** Number of consecutive assessments (and percentage values in brackets) **indicating the same direction of asymmetry** (**“no switch”**: two consecutive negative (left) or two consecutive positive (right)) or **indicating a “switch” in asymmetry direction** (negative (left) followed by positive (right) or positive (right) followed by negative (left)). Note: zero values have been considered the same “direction” as the previous assessment. Data is based on N = 1376 repeat assessments.

	HDmin	HDmax	HDup	AnyHead	PDmin	PDmax	PDup	HHD	AnyPelvis
no switch	963 (69.99)	914 (66.42)	907 (65.92)	562 (40.84)	1002 (72.82)	996 (72.38)	946 (68.75)	951 (69.11)	607 (44.11)
switch	413 (30.01)	462 (33.58)	469 (34.08)	814 (59.16)	374 (27.18)	380 (27.62)	430 (31.25)	425 (30.89)	769 (55.89)

**Table 7 animals-15-02449-t007:** Number of consecutive assessments (and percentage values in brackets) indicating a switch between a left-sided asymmetry and a right-sided asymmetry, **based on guideline values for detecting lameness,** defined for head and pelvic movement symmetry variables. Threshold values of outside +/−8 mm were used for head movement and outside +/−4 mm for pelvic movement for the definition of left and right asymmetrical movement. Data is based on N = 1376 repeat assessments.

	HDmin	HDmax	HDup	AnyHead	PDmin	PDmax	PDup	HHD	AnyPelvis
no switch	1289 (93.68)	1325 (96.29)	1229 (89.32)	1179 (85.68)	1274 (92.59)	1250 (90.84)	1151 (83.65)	1138 (82.70)	995 (72.31)
switch LR	87 (6.32)	51 (3.71)	147 (10.68)	197 (14.32)	102 (7.41)	126 (9.16)	225 (16.35)	238 (17.30)	381 (27.69)

**Table 8 animals-15-02449-t008:** Number of consecutive assessments (and percentage values in brackets) indicating a switch between a left-sided asymmetry and a right-sided asymmetry, **based on published values for daily variability in Thoroughbred racehorses** [9]. Here we chose values of +/−16.5 mm for head movement (average of range containing 90% and 95% of repeat values for HDmin and HDmax from [9]) and 10.5 mm for pelvic movement (average of range containing 90% and 95% of repeat values for PDmin and PDmax from [9]). Data is based on N = 1376 repeat assessments.

	HDmin	HDmax	HDup	AnyHead	PDmin	PDmax	PDup	HHD	AnyPelvis
no switch	1362 (98.98)	1372 (99.71)	1327 (96.44)	1322 (96.08)	1368 (99.42)	1361 (98.91)	1311 (95.28)	1280 (93.02)	1262 (91.72)
switch LR	14 (1.02)	4 (0.29)	49 (3.56)	54 (3.92)	8 (0.58)	15 (1.09)	65 (4.72)	96 (6.98)	114 (8.28)

## Data Availability

A spreadsheet with the movement symmetry difference data will be made available via figshare once the manuscript has been accepted for publication.

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
