# Peer review of "Exploring Monthly Variation of Gait Asymmetry During In-Hand Trot in Thoroughbred Racehorses in Race Training"

_animals, 2025, doi:10.3390/ani15162449_

Round 1

Reviewer 1 Report

Comments and Suggestions for Authors

Thank you for your interesting and informative submission.  I have very few comments that require any amendments. The results section was complex though, but I appreciate that there were limitations to the analysis you could apply due to the factors you highlighted within the limitations section. It may be useful to add a few more suggestions when talking about future studies on how these other factors could be gathered e.g. would a standardised data collection form or an online version help collate these data?

Line 27 – should this be “how” instead of “wow”?

Line 79 – I’m not sure that the direct question here reads well, maybe consider wording it to not be a direct question?

Line 95 – It may be useful to add that 286 horses were initially recruited for the study. I see later that you only got one measure for 30 horses which explains the different number reported in the abstract, but by making it clear that 286 was your initial population at this stage explains the difference in numbers earlier in the methods.

Line 102 – Three of unknown sex?

Line 174 – 176 – A more detailed explanation of this process would be beneficial here.

Author Response

General comments (applicable to the manuscript as a whole)

  • First, while working on the revision of the manuscript and implementing additional statistical testing, we noticed that the summary tables (6, 7, 8) had been created on a wrong version of the data set and we have revised these tables accordingly. We sincerely apologize for this oversight. The other tables and analyses were not affected by this. Neither have been the overall conclusions: there is still a very consistent reduction in ‘left-right’ switches from using “perfect symmetry” to using “lameness thresholds” and finally applying discipline-specific “repeatability values” (confirmed by the newly implemented statistical comparison).
  • Please see below for detailed responses to the individual and general comments of each reviewer. We sincerely thank the reviewers for their efforts in reviewing the manuscript. We have implemented a statistical comparison between the proportions of ‘left-right’ switches resulting from different ‘thresholds’ and have emphasized the explorative (and non-longitudinal) nature of our analysis throughout and emphasized the absence of clinical records about injuries as well as the absence of additional data about training status (and history) of the horses, simply “exploring” the frequency of more or less exacerbated ‘left-right’ switches of Thoroughbred racehorses without any further knowledge about each horse.

Reviewer 1 comments and responses: (responses in bold type face)

Thank you for your interesting and informative submission.  I have very few comments that require any amendments. The results section was complex though, but I appreciate that there were limitations to the analysis you could apply due to the factors you highlighted within the limitations section. It may be useful to add a few more suggestions when talking about future studies on how these other factors could be gathered e.g. would a standardised data collection form or an online version help collate these data?

Thank you for this suggestion. We have emphasized the need for collecting these additional data and further emphasized the lack of such training or injury data in the current study.

Line 27 – should this be “how” instead of “wow”?

Yes indeed, sorry about this mistake.

Line 79 – I’m not sure that the direct question here reads well, maybe consider wording it to not be a direct question?

Thank you for the suggestion. We have reworded this sentence.

Line 95 – It may be useful to add that 286 horses were initially recruited for the study. I see later that you only got one measure for 30 horses which explains the different number reported in the abstract, but by making it clear that 286 was your initial population at this stage explains the difference in numbers earlier in the methods.

We have amended accordingly.

Line 102 – Three of unknown sex?

Yes, due to incomplete notes and not possible to identify the horses from the online data base. Apologies.

Line 174 – 176 – A more detailed explanation of this process would be beneficial here.

Thank you for your comment. We have provided more context.

Reviewer 2 Report

Comments and Suggestions for Authors

General Comments:
This manuscript describes a large, well-executed observational study that assessed monthly variations in head and pelvic movement symmetry in 256 Thoroughbred racehorses undergoing regular training. The use of inertial sensors and standardized data acquisition over an extended period are clear strengths. The authors provide an informative exploration of how often horses switch between left- and right-sided asymmetries using various clinically and biomechanically relevant thresholds.
However, despite the robustness of the data, the manuscript relies almost exclusively on descriptive statistics and switch frequencies. The study falls short by not employing more advanced longitudinal statistical approaches, which would have significantly strengthened the interpretability and clinical impact of the findings. Additionally, while the authors suggest that subtle symmetry variations may precede injury or overuse syndromes, they do not analyze or model any clinical outcomes.
In its current form, the study has exploratory value, but it does not fulfill its potential as a predictive or inferential investigation. I encourage the authors to consider the following points.

Major concerns:

1. Lack of Longitudinal Statistical Modeling
The data structure, which includes repeated measures across time for each horse, strongly supports the use of generalized linear mixed models (GLMMs). These models could have been used to:
    - Quantify within-horse and between-horse variation.
    - Model symmetry trajectories over time.
    - Evaluate the interaction between time and asymmetry direction or magnitude.
    - Include random effects for horse, trainer, or other sources of nested variability.

2.  By relying solely on pairwise differences between time points and categorization by threshold crossing, the authors overlook the richness of the longitudinal dataset.

3. Absence of Clinical Outcome Modeling: The manuscript suggests that early detection of subtle movement asymmetries might help predict injury or poor performance. However, no clinical outcomes, such as injury occurrence, training interruption, or retirement, are included. If such data had been collected or were available, a regression analysis (e.g., logistic regression or a Cox proportional hazards model) could have been performed to determine if horses with more frequent or pronounced switches were more likely to experience injury.
If outcome data were unavailable, this should be clearly stated as a limitation.

4. Descriptive Emphasis Without Statistical Testing
The study presents extensive descriptive data (e.g., percentiles, histograms), yet no statistical hypothesis testing is performed. For instance:
    Are the observed differences between monthly and daily repeatability values statistically significant?
    Do different symmetry metrics behave differently over time?

5. Even simple inferential analyses (e.g., nonparametric tests or bootstrapped confidence intervals) would strengthen the claims.

6. No adjustment for confounding or covariates: The study does not explore whether factors such as age, sex, number of training sessions, or training intensity might influence asymmetry changes or switch frequency. Including these covariates in a GLMM framework could help elucidate patterns masked in the aggregate statistics.

Minor Comments:
- The introduction should clearly distinguish between lameness detection, injury risk assessment, and performance monitoring because these are conceptually distinct tasks.

- The methods section could be improved by including a rationale for selecting specific thresholds (e.g., 8 mm and 16.5 mm) and by including sensitivity analyses or justifications beyond previous publications.
- Some minor typographical errors remain (e.g., "how often" in the abstract).
- The Discussion is overly long and could be streamlined by moving parts of the interpretation into the Results section.

Author Response

General comments applicable to the manuscript as a whole:

  • First, while working on the revision of the manuscript and implementing additional statistical testing, we noticed that the summary tables (6, 7, 8) had been created on a wrong version of the data set and we have revised these tables accordingly. We sincerely apologize for this oversight. The other tables and analyses were not affected by this. Neither have been the overall conclusions: there is still a very consistent reduction in ‘left-right’ switches from using “perfect symmetry” to using “lameness thresholds” and finally applying discipline-specific “repeatability values” (confirmed by the newly implemented statistical comparison).
  • Please see below for detailed responses to the individual and general comments of each reviewer. We sincerely thank the reviewers for their efforts in reviewing the manuscript. We have implemented a statistical comparison between the proportions of ‘left-right’ switches resulting from different ‘thresholds’ and have emphasized the explorative (and non-longitudinal) nature of our analysis throughout and emphasized the absence of clinical records about injuries as well as the absence of additional data about training status (and history) of the horses, simply “exploring” the frequency of more or less exacerbated ‘left-right’ switches of Thoroughbred racehorses without any further knowledge about each horse.

responses to comments by Reviewer 2: (responses in bold type face)

General Comments:
This manuscript describes a large, well-executed observational study that assessed monthly variations in head and pelvic movement symmetry in 256 Thoroughbred racehorses undergoing regular training. The use of inertial sensors and standardized data acquisition over an extended period are clear strengths. The authors provide an informative exploration of how often horses switch between left- and right-sided asymmetries using various clinically and biomechanically relevant thresholds.
However, despite the robustness of the data, the manuscript relies almost exclusively on descriptive statistics and switch frequencies. The study falls short by not employing more advanced longitudinal statistical approaches, which would have significantly strengthened the interpretability and clinical impact of the findings. Additionally, while the authors suggest that subtle symmetry variations may precede injury or overuse syndromes, they do not analyze or model any clinical outcomes.
In its current form, the study has exploratory value, but it does not fulfill its potential as a predictive or inferential investigation. I encourage the authors to consider the following points.

Thank you for your thorough and constructive evaluation of our manuscript. We have addressed some of the identified issues by, as suggested below, adding some explicit hypothesis testing with regards to the number of ‘left-right switches’ identified when applying different threshold levels. We have also added the word ‘exploring’ to the title of the manuscript to emphasize the largely explorative nature of the manuscript.

We appreciate the suggestion of adding in ‘longitudinal modeling’ into our data analysis. However, after some deliberations, we have decided not to implement this type of modeling into the current manuscript. While undoubtedly these types of approaches are essential in the context of an automated, data-driven approach to ‘predicting injuries’, this is not the topic of our manuscript. Our aim (and we have clarified this throughout the manuscript and have also clarified this in the limitations section) is to explore the practical impact of utilizing gait asymmetry assessments for a ’pre-screening’ approach: i.e. what percentage of horses would have to undergo further assessments when these specific, evidence-based thresholds (for “lameness’ exams or derived from “discipline-specific repeatability” studies) were to be applied for identifying horses for further veterinary checks.

Furthermore, the data set of our current study does not provide a ‘reference event’ such as for example in previously published studies the occurrence of an injury or a period of rest or the retirement from racing. While such events will have happened for some of the horses in our study, we feel that there is no ‘common denominator’ that would allows us to implement a meaningful longitudinal modeling based on our data set.

We hope that the new version of the manuscript is now clearer and that we are no longer giving the impression that our approach is aiming at an automated, data-driven (longitudinal) prediction. Please see below for additional responses.

Major concerns:

  1. Lack of Longitudinal Statistical Modeling
    The data structure, which includes repeated measures across time for each horse, strongly supports the use of generalized linear mixed models (GLMMs). These models could have been used to:
      - Quantify within-horse and between-horse variation.
      - Model symmetry trajectories over time.
        - Evaluate the interaction between time and asymmetry direction or magnitude.
        - Include random effects for horse, trainer, or other sources of nested variability.

Please see above for some explanations. We have chosen to focus on ‘switch frequencies’ rather than implementing more complex (longitudinal) statistical modeling. We have also implemented some explicit statistical hypothesis testing and have amended the manuscript to make it clear throughout that this is the focus of the manuscript.

  1.  By relying solely on pairwise differences between time points and categorization by threshold crossing, the authors overlook the richness of the longitudinal dataset.

Please see above. Our ‘model’ is simplistic and focuses on the frequency of left-right switches in gait asymmetry without requiring the availability of a longitudinal data set with a ‘reference event’ (injury, rest, retirement) for each horse. This is comparatively simplistic and we have consequently added the word ‘exploring’ to the title of the manuscript.

  1. Absence of Clinical Outcome Modeling: The manuscript suggests that early detection of subtle movement asymmetries might help predict injury or poor performance. However, no clinical outcomes, such as injury occurrence, training interruption, or retirement, are included. If such data had been collected or were available, a regression analysis (e.g., logistic regression or a Cox proportional hazards model) could have been performed to determine if horses with more frequent or pronounced switches were more likely to experience injury.
If outcome data were unavailable, this should be clearly stated as a limitation.

There is no clinical data available for the horses, and we have made this more explicit in the limitations section and are emphasizing the focus on using gait asymmetry assessments as a ‘pre-screening’ tool as opposed to a longitudinal ‘prediction’ approach. We have added the word ‘exploring’ to the title of the manuscript.

  1. Descriptive Emphasis Without Statistical Testing
    The study presents extensive descriptive data (e.g., percentiles, histograms), yet no statistical hypothesis testing is performed. For instance:
      Are the observed differences between monthly and daily repeatability values statistically significant?

We have implemented a statistical test to evaluate whether the ‘switch frequencies’ are statistically significantly different between the “lameness” and the “repeatability” thresholds as well as in comparison to using “perfect symmetry” as the switch threshold.
    Do different symmetry metrics behave differently over time?

Please see comments above about the decision not to implement any longitudinal monitoring.

  1. Even simple inferential analyses (e.g., nonparametric tests or bootstrapped confidence intervals) would strengthen the claims.

Please see above for statistical testing implemented between the different thresholds in terms of switch frequency.

  1. No adjustment for confounding or covariates: The study does not explore whether factors such as age, sex, number of training sessions, or training intensity might influence asymmetry changes or switch frequency. Including these covariates in a GLMM framework could help elucidate patterns masked in the aggregate statistics.

Please see above.

Minor Comments:
- The introduction should clearly distinguish between lameness detection, injury risk assessment, and performance monitoring because these are conceptually distinct tasks.

Thank you. We have added a little more detail to the relevant section in the introduction. The new section now reads:

“We are interested in further evaluating whether such quantitative movement symmetry data might be useful for exploring the onset of impending overuse injuries, for example by identifying horses that may benefit from further veterinary assessments. Importantly, this task is different from clinical lameness examinations, i.e. it is different from identifying the lame(st) limb in a horse with a pre-conceived notion that “some-thing is wrong” and further clinical examinations need to be conducted. In the context of lameness examinations, the measurement of (small amounts of) movement asymmetry is used for identifying a limb that is contributing less to force production in comparison to its contra-lateral counterpart. In the context of detecting impending injuries, the task is to identify horses that are likely to develop an injury in the future and for example have been shown to show comparatively small adaptations to their “in-race” gallop stride parameters [14]. We are asking the question, whether longer term (monthly) variations in movement symmetry might be useful in this context by quantifying the frequency of switches between left and right-sided movement asymmetries as a ‘pre-screening’ tool for identifying horses that might benefit from further veterinary assessments. We implicitly assume that switches between more pronounced movement asymmetries, which have been related to larger force asymmetries [12,13], are more likely related to an attempt of ‘unloading’ one of the limbs, and we postulate that such horses might benefit from further veterinary inspections.”

- The methods section could be improved by including a rationale for selecting specific thresholds (e.g., 8 mm and 16.5 mm) and by including sensitivity analyses or justifications beyond previous publications.

Thank you. We have added some explanations about the chosen “lameness” threshold. The focus of our manuscript is on providing an estimation of the number (percentage) of gait assessments that would ‘trigger’ further ‘pre-screening’ assessments. We are emphasizing the use of a ‘universally’ accepted ‘lameness’ threshold compared to a ‘discipline-specific’ variability threshold both based on previously published evidence.

- Some minor typographical errors remain (e.g., "how often" in the abstract).

Thank you. We have corrected this.

- The Discussion is overly long and could be streamlined by moving parts of the interpretation into the Results section.

Thank you. We have removed some content from the discussion that is a little less directly relevant for the main focus of the study: switch frequencies between left and right sided gait asymmetry. Given that the results section is already somewhat long, we would like to leave the ‘interpretation’ in the discussion section and we have shortened the discussion by removing less directly relevant discussion points.

Round 2

Reviewer 2 Report

Comments and Suggestions for Authors

I would like to thank the authors for their detailed and thoughtful responses, and for their efforts in revising the manuscript. They addressed several key points, such as clarifying the study's exploratory nature, performing basic hypothesis testing regarding switch frequency between thresholds, and correcting errors related to generating the data table. These changes have improved the transparency and clarity of the manuscript.

I particularly appreciate the decision to revise the title to emphasize the study's non-inferential and descriptive intent. The distinction between using gait asymmetry as a clinical lameness diagnostic tool versus a potential pre-screening signal for further veterinary evaluation is clearer now. Including statistical testing for differences in switch frequency between symmetry thresholds also represents a meaningful improvement over the original version.

However, some limitations remain. The decision not to implement longitudinal statistical modeling (e.g., generalized linear mixed models [GLMMs]), despite the repeated-measures structure of the dataset, is methodologically conservative. While the authors justify this choice based on missing clinical outcome data, basic modeling of within-horse variation or time trends would have provided valuable insights without requiring injury or retirement data. Nevertheless, the authors' explicit framing of the study as exploratory mitigates this concern.

Overall, the revised manuscript more clearly aligns with its scope and limitations. Although its clinical applicability is modest due to the lack of outcome data and the descriptive nature of the analysis, the study contributes useful groundwork to the field of movement symmetry monitoring and may stimulate further research with stronger predictive objectives.